# Sustained Energy Deficit Following Perinatal Asphyxia: A Shift towards the Fructose-2,6-bisphosphatase (TIGAR)-Dependent Pentose Phosphate Pathway and Postnatal Development

**DOI:** 10.3390/antiox11010074

**Published:** 2021-12-29

**Authors:** Carolyne Lespay-Rebolledo, Andrea Tapia-Bustos, Ronald Perez-Lobos, Valentina Vio, Emmanuel Casanova-Ortiz, Nancy Farfan-Troncoso, Marta Zamorano-Cataldo, Martina Redel-Villarroel, Fernando Ezquer, Maria Elena Quintanilla, Yedy Israel, Paola Morales, Mario Herrera-Marschitz

**Affiliations:** 1Molecular & Clinical Pharmacology Program, ICBM, Faculty of Medicine, University of Chile, Santiago 8380453, Chile; carolynelespay@gmail.com (C.L.-R.); ronald.perezlobos@gmail.com (R.P.-L.); valeviomunoz@gmail.com (V.V.); emmanuel.casanovao@gmail.com (E.C.-O.); nancy.farfan.t@gmail.com (N.F.-T.); zamorano.c.marta@gmail.com (M.Z.-C.); martina.redel90@gmail.com (M.R.-V.); equintanilla@med.uchile.cl (M.E.Q.); yisrael@uchile.cl (Y.I.); 2School of Pharmacy, Faculty of Medicine, Universidad Andres Bello, Santiago 8370149, Chile; ac.tapiabustos@gmail.com; 3Center for Regenerative Medicine, Faculty of Medicine-Clínica Alemana, Universidad del Desarrollo, Santiago 7710162, Chile; eezquer@udd.cl; 4Department of Neuroscience, Faculty of Medicine, University of Chile, Santiago 8380453, Chile

**Keywords:** hypoxia, brain plasticity, redox homeostasis, basal ganglia, hippocampus, catalase, caspase, pentose-phosphate-pathway, organotypic cultures, mesenchymal stem cell secretomes, rat

## Abstract

Labor and delivery entail a complex and sequential metabolic and physiologic cascade, culminating in most circumstances in successful childbirth, although delivery can be a risky episode if oxygen supply is interrupted, resulting in perinatal asphyxia (PA). PA causes an energy failure, leading to cell dysfunction and death if re-oxygenation is not promptly restored. PA is associated with long-term effects, challenging the ability of the brain to cope with stressors occurring along with life. We review here relevant targets responsible for metabolic cascades linked to neurodevelopmental impairments, that we have identified with a model of global PA in rats. Severe PA induces a sustained effect on redox homeostasis, increasing oxidative stress, decreasing metabolic and tissue antioxidant capacity in vulnerable brain regions, which remains weeks after the insult. Catalase activity is decreased in mesencephalon and hippocampus from PA-exposed (AS), compared to control neonates (CS), in parallel with increased cleaved caspase-3 levels, associated with decreased glutathione reductase and glutathione peroxidase activity, a shift towards the TIGAR-dependent pentose phosphate pathway, and delayed calpain-dependent cell death. The brain damage continues long after the re-oxygenation period, extending for weeks after PA, affecting neurons and glial cells, including myelination in grey and white matter. The resulting vulnerability was investigated with organotypic cultures built from AS and CS rat newborns, showing that substantia nigra TH-dopamine-positive cells from AS were more vulnerable to 1 mM of H_2_O_2_ than those from CS animals. Several therapeutic strategies are discussed, including hypothermia; N-acetylcysteine; memantine; nicotinamide, and intranasally administered mesenchymal stem cell secretomes, promising clinical translation.

## 1. Pregnancy and Delivery

In humans, pregnancy lasts 9 months, a period that is divided into three trimesters: the first trimester, from conception up to the time when the embryo and placenta are formed; the second trimester, when the movements of the fetus may already be felt, and the third trimester, starting at week 28, culminating with childbirth. At week 28, the babies can survive outside of the uterus if properly supported. A delivery is considered as *preterm* when occurring before 37 weeks, *on the term* when occurring at 37–41 weeks, and *post-term* when occurring beyond 41 weeks of gestation.

Pregnancy culminates at the time when labor begins, entailing a complex interchange of molecules generated by uterine and extrauterine tissue. There is an increase in myometrial contractility, cervical dilatation, decidual/membrane activation and rupture of chorioamniotic membranes [1], and a shift from anti-inflammatory (e.g., IL-4; IL-6: IL-10) to pro-inflammatory (e.g., IL-1β, 12, 18, TNFα, IFN-γ) signaling cytokines and chemokines [1], as well as contraction-associated molecules (e.g., estrogen, connexins, prostaglandins, oxytocin). Progesterone maintains uterine quiescence, opposing the effects of estrogen, but in the near term, progesterone can enhance the effects of estrogen or, by other mechanisms, increase coordinated rhythmic contractions that lead to a triple descending gradient [2]. Cervical ripening stimulates cervical dilatation, widened by the repeated uterine contractions. Thus, labor and delivery involve a complex and sequential metabolic and physiologic cascade, culminating in most circumstances in successful childbirth. Nevertheless, delivery can be a risky event, mainly when there is a metabolic and/or a blood perfusion deficit, including a drop in maternal blood pressure during labor, preceding delivery, or during and/or after delivery, when the required spontaneous pulmonary respiration by the newborn is delayed or interrupted, leading to reduced transfer of oxygen to tissue and cells, and hypoxia, either due to a dysfunction of respiratory and heart rate by the mother at the prenatal stage, or by the newborn after delivery, resulting in a drop of oxygen saturation.

Interruption of oxygen transference leads to perinatal asphyxia (PA), a medical condition occurring at labor, delivery, and/or neonatal stages, affecting most of the infant’s organs, but the brain being the most concerned target. The incidence of PA is high; 2 to 10 per 1000 on term neonates [3], while is higher among premature deliveries [4,5,6]. Furthermore, the incidence of PA can be 5 times higher in developing- than in developed-countries [7], largely associated with prolonged labor and insufficient medical care [3]. If re-oxygenation is re-established, PA can lead to long-term consequences [8], including long-lasting neuropsychiatric dysfunctions when children reach critical developmental stages [9,10].

## 2. The Pathophysiological Cascade Elicited by PA

PA can induce long-term biological vulnerability depending on its severity, independent of any genetic or clinical predisposition [11,12]. PA (an environmentally dependent variable) refers to an unexpected oxygen interruption at the time of labor, or when delivery has already begun. Therefore, no genetic factor, malformation, or prematurity is required for the clinical entity of PA, which is defined as a specific metabolic/energetic insult related, first, to delay and/or interruption of autonomous breathing and, second, to re-oxygenation, a requirement for survival. PA provides a framework to address a fundamental issue affecting long-term CNS plasticity. PA produces metabolic alterations affecting the maturity of neuronal networks, including neurocircuitries of basal ganglia and hippocampus [13,14,15], impairing the ability of the central nervous system (CNS) to cope with stressors occurring during life [16,17]. Thus, the perinatal insult triggers a domino-like sequence of events making the developing individual vulnerable to recurrent adverse conditions, decreasing his/her coping repertoire because of a relevant insult occurring at birth [16].

### 2.1. Free Radical Reactive Species

Gas exchange dysregulation results in hypoxemia, hypercapnia, and metabolic acidosis of vital organs, including the brain [18], producing energy failure, and a biochemical cascade leading to cell dysfunction and ultimately to cell death. Hypoxia leads to the generation of reactive oxygen (ROS) and nitrogen (RNS) species, which inhibit prolyl-hydroxylases that under normoxia metabolize the oxygen sensor Hypoxia Inducible Factor-1alpha (HIF-1α), to be poly-ubiquitinated by the von Hippel-Lindau tumor-suppressing factor (pVHL), and eliminated by the proteasome [19], a discovery that led to the Nobel Prize of Physiology & Medicine 2019 (Gregg Semenza, William Kaelin, and Peter Ratcliffe). Following the interruption of oxygen availability, HIF-1α accumulates and translocates to the nucleus, stimulating the expression of its transcriptional targets via binding to hypoxia-responsive elements (HREs), which are present in genes associated with cell metabolism and mitochondrial function. The tricarboxylic acid cycle is down-regulated, while anaerobic glycolysis is up-regulated, to allow the cells to cope with the low oxygen tension [20,21]. Translocation of HIF-1α stimulates pro-apoptotic genes, including the Bcl-2 family members Nix, Noxa, Bnip3, and Apoptosis-Inducing Factor (AIF) [22,23], also the expression of sentinel proteins, *vide infra* [24].

### 2.2. A Switch to Anaerobic Glycolysis

During hypoxia, there is a switch to anaerobic glycolysis, for neurons a poor alternative, because of a low store of glucose in the immature brain [25], and a low yield of adenosine triphosphate (ATP). During glycolysis one molecule of glucose breaks down into two molecules of pyruvate, fermenting to lactate when oxygen is lacking, accumulating in intracellular and extracellular compartments, resulting in a decrease of pH, and acidosis [26].

### 2.3. Glutamate and Extrasynaptic Glutamate Receptors

The weakening of ATP-dependent transport pumps impairs the uptake of neurotransmitters, mainly that of glutamate, largely synthesized by the astroglial-neuronal glutamine shuttle. Extracellular glutamate levels, buffered by Na^+^/K^+^-ATPase-dependent transporters, are taken up by glial and neuronal cells for metabolic degradation and/or re-cycling [27,28,29]. Free radicals also affect glutamate transport by a sulfhydryl-based regulatory mechanism, making the transport sensitive to redox agents [28,30]. Under sustained hypoxia the half-life of extracellular glutamate is prolonged, providing an extreme homeostatic response for removing the organism from a catastrophic condition, generating widespread neuronal depolarization by extracellular glutamate binding to any available glutamate receptor, mainly of the extrasynaptic subtype. Indeed, at birth, extrasynaptic GluN2B-containing N-methyl-D-aspartate receptors (NMDARs) prevail, associated with excitotoxic cascades, seizures, and cell death [31,32], via increased calcium conductance and massive mitochondrial loading [33]. Over-stimulation of extrasynaptic NMDARs increases nitric oxide (NO) production and further free-radical stress by peroxynitrite and its reaction with superoxide anions. NO can decrease mitochondrial membrane potentials, liberating pro-apoptotic proteins [34], including AIF, NADPH oxidase, and neuronal nitric oxide synthase (nNOS) [35], inducing DNA fragmentation and mitochondrial fission, maintaining a condition of high ADP/ATP ratio and energy inefficiency [36,37].

### 2.4. Mitochondrion: A Main Actor and a Vulnerable Target

Permeabilization of the outer membrane of the mitochondria (MOMP) releases cytochrome C and/or Smac/Diablo proteins from the mitochondrial matrix, translocated to the cytosol and nucleus, activating caspase-dependent apoptosis [38]. The NAD^+^ cofactor leaks via the inner membrane permeability transition pore (PTP), further increasing intracellular calcium and oxidative stress, leading to overactivation of poly(ADP-ribose) polymerases (PARPs), also activated by DNA modifications [39].

### 2.5. Sentinel Proteins

Suppression and/or overactivation of gene expression occur immediately during hypoxia or during the re-oxygenation period following PA. When DNA integrity is compromised, sentinel proteins are activated, including PARPs, X-ray Cross complementing Factor 1 (XRCC1), DNA ligase IIIα, DNA polymerase β, Excision Repair Cross-Complementing Rodent Repair Group 2 (ERCC2), and DNA-dependent protein kinases, all of them shown to play a role, diminishing or worsening the effect of hypoxia [40].

#### 2.5.1. Poly(ADP-ribose) Polymerases

The PARPs bind to glutamic acid residues on acceptor proteins, inducing post-translational modifications. The (ADP)r polymers are linked by glycoside ribose-ribose bonds, forming branched structures [41]. The synthesis of pADPr requires three distinctly enzymatically-regulated activities: (i) mono(ADP-ribosyl)ation of the substrate; (ii) elongation, and (iii) branching of the polymers. The principal and most abundant member of the PARP superfamily, PARP-1, possesses all three activities [42], therefore responsible for the majority of the poly(ADP-ribosyl)ation in living cells [43].

PARP-1 is a protein formed by four domains: (i) The DNA-binding domain (DBD), located on the N-terminal region, containing two zinc fingers, responsible for the binding to DNA breaks [44]; (ii) a bipartite nuclear localization signaling domain (NLD) for homing PARP-1 to a caspase-3 cleaved site [45]; (iii) an auto modification domain containing a BRCAI C-terminal motif (BRCT; an amino-terminal zinc finger motif, described in breast cancer tumors) in the center of the sequence, a conserved protein-protein interaction domain, involved in DNA repair, recombination and cell cycle control [46], and (iv) the smallest PARP-1 fragment, retaining catalytic activity on the C-terminal region (the PARP signature) [47]. PARP-1 signaling occurs via the attachment of ADP-ribose chains to nuclear proteins recognized by DNA-repairing enzymes, such as DNA ligase III [48]. PARP-1 uses NAD^+^ to synthesize linear or multi-branded polymers of ADP-ribose, to be linked to a large number of protein acceptors associated with chromatin [49,50]. The generation of ADP-ribose monomers requires NAD^+^, which is why PARP-1 overactivation further depletes NAD^+^ stores, resulting in progressive ATP depletion [51,52]. Furthermore, there is tight crosstalk between PARP-1 and HIF-1α [53]. Under hypoxic conditions and/or oxidative stress, PARP-1 modulates HIF-1α activity, preventing its proteasome degradation [54].

#### 2.5.2. X-ray Repair Cross Complementing 1 (XRCC1) Protein

DNA strand breaks lead to polyADP-ribosylation of sites that accumulate XRCC1 [55,56], a scaffold protein forming part of the DNA base excision repair (BER) pathway, associated with DNA fragmentation and apoptosis, as well as with necrosis-dependent cell death. XRCC1 is thought to interact with DNA ligase III by its BRCT domain, which is a widespread motif for DNA repair, also interacting with DNA polymerase β. Chiappe-Gutierrez et al. (1998) [57] reported a 5-fold increase in XRCC1 levels 30 min after global hypoxia, reaching a maximal expression at 4 h. Nevertheless, Fujimura et al. (1999) [58] found that XRCC1 levels were decreased after transient focal ischemia, preceding apoptosis. A discrepancy that should be further investigated.

### 2.6. A Schematic Summary of the Metabolic Cascade Elicited by PA

Figure 1 summarizes the pathophysiological cascade elicited by PA, describing several phases leading to short- and long-term brain deficits. A primary phase relates to the length of hypoxia, implying energy deficit, oxidative stress, and cell death. Re-oxygenation is a requisite for survival, leading to a latent phase, characterized by oxidative stress, excitotoxicity, epileptiform transients (characterized by small sharp spikes, SSSs), neuroinflammation, and pro-apoptotic signaling. In a secondary phase, the oxidative stress and the neuroinflammatory cascades overpass the mitochondrial capacity to sustain the increased metabolic demands, collapsing, and leading to excitotoxicity and seizures. At a tertiary phase, sustained oxidative stress and inflammatory cascades result in regionally specific cell death, impaired connectivity, and altered maturation, making the neonate vulnerable to metabolic insults later in life.

## 3. An Experimental Model of Global PA in Rats

The issue of PA has been studied in rodents, and larger animals, such as sheep and the South American Lama Glama (llama), summarized by excellent reviews [59,60,61].

In the present paper, we summarize results obtained with an experimental model of global PA performed in Sprague-Dawley rats, originally proposed by Börje Bjelke, Kurt Andersson and collaborators at the Karolinska Institutet, Stockholm, Sweden, in the early 1990s [62,63,64], established now in our laboratory at the University of Chile, Santiago, Chile, using Wistar rats, finding no relevant differences among the rat strains (see Figure 2). In this model (see Figure 1) hypoxia occurs at the time of delivery. The model has been useful for identifying relevant targets responsible for the metabolic cascades leading to short- and long-term effects, evaluated in vivo and in vitro [40,59,65].

The experiments start by evaluating the estrus cycle in a vaginal smear of a young female rat (about 200 g, approximately one month and half of the age), to be exposed to a male for one night at the time of the pro-estrus. Thereafter, the presence of sperm and a vaginal clot is evaluated to predict the exact time of delivery (22 days). Clinical and behavioral observations are recorded up until the spontaneous delivery of a first fetus. The dam is then euthanized by neck dislocation and subjected to cesarean section and hysterectomy. The fetuses containing uterine horns are immediately immersed into a water bath at 37 °C for various periods of time (0–22 min), thereafter the pups are removed from the uterine horns. The pups are then resuscitated by cleaning the faces of the animals from fluid and amniotic tissue, freeing the mouth and the nose to stimulate pulmonary breathing. Pups exposed to cesarean delivery only (CS, 0 min of asphyxia) start breathing as soon as the mouth and the nose are cleaned from amniotic tissue. For pups exposed to severe asphyxia (AS, 21 min of asphyxia), resuscitation requests of expert and skillful handling, taking several minutes before the pup produces the first gasping, and even longer time for more-or-less regular breathing, always supported by gasping. 40 min after delivery (accounted from the time the uterine horns are excised from the abdominal cavity, cutting the uterine artery and venous at the level of the cervix), surviving pups are evaluated with an Apgar scale adapted to rats, monitoring the body weight, sex, color of the skin, respiratory frequency, gasping, vocalization, muscular rigidity and spontaneous movements [66,67]. The evaluation can be extended to several days after delivery [68,69]. The Apgar evaluation is critical for this experimental model because it assesses the extent of the insult since severity is directly correlated with the rate of survival and recovery by the pups of a litter. A 100% survival is observed whenever fetuses containing uterine horns are immersed for up to 15 min in a water bath at 37 °C. The rate of survival drops rapidly until no survival is observed following 22 min of asphyxia. The recovery period can prolong the hypoxic condition since surviving pups show a decreased breathing rate, decreased cardiovascular function, and blood perfusion [69]. The Apgar provides information about the cesarean-delivered control pups, whose physiological parameters have to be similar to those shown by vaginally delivered pups. The obtained parameters are useful to compare results obtained by different labs and/or different treatments. The handling of the pups is also important for their reception by a surrogate dam (selected from the same cohort of pregnant rats, delivering just before the dam chosen for a cesarean section, the asphyxia-exposed and cesarean delivered pups replacing those delivered by the surrogate dam).

The model and the experiments have been regularly evaluated for bioethics by the Local Ethics Committee for experimentation with laboratory animals at the Medical Faculty, the University of Chile (the latest being Protocol CBA#1058 FMUCH, 19255-MED-UCH; approval date: 6 May 2019), and by an ad-hoc commission of the Chilean Council for Science and Technology Research (recently, FONDECYT#1190562, approval date 4 March 2019), endorsing the principles of laboratory animal care. Animals are permanently monitored (on daily basis) regarding their wellbeing, following the ARRIVE guidelines for reporting animal studies (www.nc3rs.org.uk/arrive, accessed on 20 March 2021).

The present experimental model of global PA in rats allows to monitor early or delayed long-lasting molecular, metabolic, and physiological effects, and the rat pups can also be used to prepare organotypic cultures [70,71]. Nevertheless, caution is advised when extrapolating to human data obtained in rodents. Compared to humans, the brain of rodent neonates is premature, based on neocortex development [72] and the pattern of the oligodendrocyte (OL) lineage progression required for cerebral myelination [68,73]. Nevertheless, despite the obvious limitations, experimental models in rodents have led to significant progress for the understanding of the mechanisms of the long-term impairments produced by PA.

## 4. The Energy Crisis Induced by Global PA

PA is a menace to the full organism, affecting systemic and brain tissue. The availability of ATP is rapidly decreased in the kidneys, already depleted after 5 min of PA, whereas in the brain ATP is decreased to less than 50% after 15 min of asphyxia if performed at 37 °C, while in the heart ATP is sustained for longer than 20 min until the lack of oxygenation is incompatible with life (22 min) [65,74,75]. Heart metabolism can be supported by the “phosphocreatine shuttle” [76], which was suggested not to be a valid metabolic pathway for the neonatal brain [77]. Further research is, however, required to evaluate the phosphocreatine shuttle in the developing brain, since there is evidence showing that a creatine-supplemented maternal diet can protect the newborn from birth hypoxia [78,79,80].

PA implies a long-term metabolic insult, triggered by the length of the induced hypoxia, the resuscitation/reoxygenation maneuvers, but also upon the developmental stage of the affected brain regions, and the integrity of cardiovascular and respiratory physiological functions [69], which are fundamental for warrantying a proper development.

### 4.1. Redox Homeostasis

Redox homeostasis is continuously dealing with metabolic challenges produced by electrophiles and nucleophiles, generating free radical species. The role of oxidative stress on cell damage has been studied in models of hypoxia/ischemia at P7, implying vessels ligation and hypoxic chambers, focusing on the reperfusion-oxygenation period [81,82,83], also following global perinatal asphyxia [84,85]. A relationship between the duration of asphyxia and ROS production has been demonstrated, both in animal and clinical studies [86], proposing that oxidative stress markers predict the severity of damage induced by PA [87,88,89,90].

We studied the effect of PA on redox homeostasis during the first weeks after birth, monitoring mesencephalon, telencephalon, and hippocampus, brain areas shown to be susceptible to hypoxia and ischemic insults [62,66,91,92]. The ratio oxidized (GSSG): reduced (GSH) glutathione was calculated as a marker of oxidative stress, based on the fact that GSH is the major non-enzymatic antioxidant regulator of redox homeostasis [93]. We observed an increase in the GSSG:GSH ratio in mesencephalon and hippocampus up to P14 [69,94,95], supporting the idea that PA induces long-term oxidative stress, impairing redox homeostasis.

### 4.2. Glutathione (GSSG:GSH Ratio) and Glutathione Reductase

GSH plays a fundamental role in keeping the balance between ROS production and antioxidant defenses [96]. The synthesis of glutathione is modulated by Nrf-2, a transcriptional factor responding to moderate ROS levels, increasing the transcription of genes coding for antioxidant enzymes of phase II metabolism [97]. ROS levels induce degradation of Nrf-2 by ubiquitination, decreasing the expression of antioxidant enzymes [98]. Glutathione reductase (GR) restores GSH levels by reducing GSSG [99,100]. The activity of GR is NADPH-dependent, hence from glucose and ATP availability; promoting the pentose phosphate pathway (PPP) for maintaining NADPH physiological levels [101]. In models of hypoxia/ischemia in postnatal rats, the PPP showed a reduced activity after the hypoxic/ischemic insult, leading to decreased NADPH levels and decreased GR activity [102,103], also implying mitochondrial hypo-metabolism [104].

We observed accumulation of GSSG in mesencephalon and hippocampus [69,94,95], associated with decreased GR activity and NADPH levels in hippocampus from asphyxia-exposed animals [94], contributing to a sustained oxidative stress, affecting the synthesis of *de novo* GSH, regulated by the Nrf-2 pathway [105], indicating that PA impairs the GR-dependent salvage pathway for GSH recycling. The delayed response on GSH levels, decreased in mesencephalon and hippocampus after PA [69], indicates a failure of *de novo* synthesis of GSH, associated with metabolic deregulation of cysteine, glycine, and/or glutamate, precursors of GSH synthesis by astrocytes [106].

The effect of PA on the GSSG:GSH ratio was supported by studies evaluating the tissue antioxidant capacity with the *Ferric Reducing Antioxidant Power Assay* (FRAP), showing a decreased antioxidant capacity after PA, both in mesencephalon and hippocampus [94]. FRAP is directly associated with the availability of reducing molecules, such as ascorbic acid, tocopherol, ubiquinone, NADPH, L-carnitine, and free thiols [107]. Hence, a decrease in FRAP levels supports the idea of a decreased synthesis of NADPH, favoring GSSG accumulation, leading to oxidative stress, observed during the first weeks after birth, explaining a delayed response on GSH levels, decreased in mesencephalon and hippocampus after PA. In the telencephalon, FRAP levels were only increased at P7, as compared to control neonates, indicating a local, instead of a generalized effect, supporting the proposal that oxidative stress depends upon the generation of free radical species, as well as upon the intrinsic defenses against the insult [69,94]. In agreement, telencephalon has been shown to be a more resilient brain region against hypoxia and ischemia that are induced at early neonatal stages [92], probably reflecting a delayed postnatal development [108].

### 4.3. Impaired Control of Peroxidation

Catalase and glutathione peroxidase (GPx) act cooperatively to control cell hydro-and lipid-peroxides levels, catalase decomposing hydrogen peroxide (H_2_O_2_) to water and molecular oxygen [109,110]. There is a direct relationship between the decomposition of H_2_O_2_ levels by catalase and resistance to oxidative stress following hypoxia-ischemia [103,111]. At high levels (>100 µM), H_2_O_2_ is detrimental to the cells, leading to activation of cell death cascades and GSH depletion [109], favoring oxidative modifications of proteins [112,113,114].

Catalase is also auto-inactivated at high concentrations of H_2_O_2_, resulting in an irreversible inhibition, which is prevented by GPx, removing H_2_O_2_, maintaining it at a low concentration (~1 µM), allowing the functioning of catalase [115,116]. However, the reduction of peroxides by GPx requires GSH for its catalytic mechanism [109,110,117].

We monitored catalase protein levels, estimating the enzymatic activity by the constant (k) from the exponential degradation of H_2_O_2_, normalized by catalase relative levels, finding that catalase activity was decreased up to P14, both in mesencephalon and hippocampus from AS, compared to CS neonates. The maximum effect was observed in the hippocampus at P14. No significant changes were observed in telencephalon [69], reflecting a delayed postnatal development. The antioxidant system of the neonatal brain matures along with development, a main factor of vulnerability compared to that shown by the adult brain [90,118,119]. It was reported that the neonatal brain (P7) accumulates more H_2_O_2_ levels than the adult brain (P42), up to 5 days after a postnatal hypoxic-ischemic insult [120]. Nevertheless, no changes in catalase activity were observed at the first hours after a hypoxic-ischemic insult induced at P7 [121], but it was increased in the adult brain [92].

In our study, however, catalase activity was decreased in mesencephalon and hippocampus, up to 14 days after PA, without changes in protein expression [69,94], suggesting that oxidative stress produced by PA may generate post-transductional modifications by nitrosylation and oxidation on catalase residues affecting its enzymatic activity without inducing changes in protein levels [122,123,124]. In the same region (mesencephalon and hippocampus) and same period (P14), there was a >2-fold increase of the GSSG:GSH ratio, indicating elevated oxidative stress [60,102]. Similar results have been reported, based on different experimental models, showing that catalase activity is decreased by oxidative stress, without changes in protein expression, also explained by post-transductional modifications [122,123].

### 4.4. TIGAR Modulation of the Pentose Phosphate Pathway

TIGAR (Tp53-inducible glycolysis and apoptosis regulator) identified as a p53-inducible gene [125,126] functions as a fructose-2,6-bisphosphatase, helping to maintain NADPH levels, which are generated by the PPP, oxidizing glucose-6-phosphate by glucose-6-phosphate dehydrogenase (G6PGH), resulting in the formation of two molecules of NADPH, which contribute to maintaining the activity of antioxidant enzymes, such as GR, a key enzyme for maintaining cell GSH levels and a basal GSSG:GSH ratio [127]. A reduced PPP flux has been reported in the brain of animals exposed to postnatal hypoxia-ischemia, correlating with a reduction of GR activity [102].

Under basal conditions, G6PGH has low enzymatic activity, limiting the oxidation of glucose-6-phosphate via the PPP, favoring glucose metabolism to produce ATP. But, under hypoxia-ischemia conditions a rapid upregulation of TIGAR is observed in neurons and astrocytes, allowing the glucose metabolism to shift to the PPP [126,128]. The fructose-2,6-bisphosphatase activity of TIGAR degrades fructose-2,6-biphosphate, inhibiting phosphofructokinase 1 (PFK1), a rate-limiting enzyme of glycolysis [125,126,129]. The inhibition of glycolysis by TIGAR results in the shunt of glucose-6-phosphate to PPP [125], enhancing its flux, generating NADPH, reducing GSSG to GSH, decreasing ROS levels [125,130]. TIGAR can re-localize to the outer mitochondrial membrane, increasing the activity of hexokinase 2 (HK2), to maintain the mitochondrial membrane potential, thus reducing ROS levels, preventing caspase-depending apoptosis [131,132].

TIGAR expression decreases along with the development. Lower levels of TIGAR are observed in the adult than in the neonatal brain [126]. In postnatal models of brain ischemia, it was observed that TIGAR was up-regulated, reaching a peak at 3 h post-reperfusion, declining thereafter toward basal levels [126,133]. Nevertheless, we found that TIGAR levels were decreased in the hippocampus from AS animals at P1, but increased at P14 [94]. The decrease of TIGAR levels observed at P1 occurred together with a decrease of GR, GPx, and catalase activity, at a time when the GSSG:GSH ratio was increased ~2-fold. At P14, TIGAR levels were increased, when the GSSG:GSH ratio increased >4-fold in AS, compared with CS animals, and GPx and catalase activity was decreased by more than 50%.

The downregulation of TIGAR observed at P1 suggests a failure of the cells to shunt glucose-6-phosphate to the PPP for producing NADPH during the postnatal period, probably because of a reduced GR activity, also observed in models of postnatal brain hypoxia/ischemia, reducing PPP in the brain from unilaterally clamped carotid arteries, correlating with reduction of GR activity 2 h after the insult [102].

The increase of TIGAR expression observed at P14, together with a reduction of GR activity, suggests a compensatory delayed mechanism, increasing the PPP. However, NADPH levels did not appear to enhance the activity of NADPH-dependent enzymes, since the oxidative stress was sustained, evidenced by a high GSSG:GSH ratio still observed at P14. It is proposed that the NADPH produced by PPP in AS animals are used to generate superoxide anions by the action of NADPH oxidase, which is supported by in vitro evidence [134,135,136,137], showing that superoxide anion production depends on glucose metabolism, via the hexose monophosphate shunt to generate NADPH after ischemia-reperfusion [138,139].

### 4.5. Delayed Cell Death as a Consequence of Long-Term Impaired Redox Homeostasis

In a model of postnatal hypoxia/ischemia, it was reported that after the insult, perfusion and oxygenation could be re-established, recovering ATP levels and cellular homeostasis [140]. However, there was a secondary ATP decreased period 8*–*16 h after the insult, which was even enhanced 24*–*48 h later, a phenomenon known as secondary energy failure [83,141]. This secondary energy failure results in delayed cell death, which is progressive and can last days, weeks even months after the insult [142,143,144,145,146].

In the global PA model reported here, delayed cell death is promoted by activation of apoptotic mechanisms, increasing molecular mediators of apoptosis and TUNEL positive cell death [17,66,91,147,148]. We observed [69] that cleaved caspase-3 levels were increased in the mesencephalon of PA neonates at P1–14, while in the hippocampus that increase was only observed at P3–7, and no effect was observed in the telencephalon. The PA-induced apoptotic cell death was sustained along with development, in parallel with a high GSSG:GSH ratio and reduced catalase activity, observed in vulnerable brain areas, in agreement with several studies showing oxidative and nitrosative stress, causing DNA fragmentation, mitochondrial dysfunction, and activation of caspase-dependent cell death [149,150]. Indeed, in models of postnatal hypoxia/ischemia [58,151], it has been shown that oxidative stress produces DNA strand breaks, a polyADP-ribosylation-dependent decrease of XRCC1 levels, apoptosis, and necrosis-dependent cell death. However, we observed an increase of XRCC1 levels in the hippocampus of AS neonates, but only at P1, without any change at P14 [94], also an increase of PARP-1 expression, restricted to the first hours after delivery [147,152].

Oxidative stress induced by PA can also lead to increased cytosolic calcium levels, mitochondrial dysfunction, and/or endoplasmic reticulum stress, resulting in the activation of cell death mediated by calpain. We observed that calpain activity increased in the hippocampus from rats exposed to PA at P14, a time at which an increase in caspase-3 levels were no longer observed, suggesting a caspase-3 independent, but bax-dependent delayed cell death [69,94]. The activation of calpain has been associated with fragmentation of dendritic processes and neuronal degeneration in models of postnatal hypoxia/ischemia [153], observing an initial activation of calpain 1, shortly after the insult, decreased at 2–8 h, but again increased at P14 to P21 [154,155], leading to apoptosis independent cell death, associated to caspase 7, 8, and 9 [156,157]. Calpains are also essential for necrosis and programmed necrosis induced by TNF-α [158,159].

Whether calpain mediates the cell damage induced by PA in the hippocampus, via necrosis or programmed necrosis, is a matter of further investigation, including evaluation of the role of AIF release and/or TNF-α signaling, as shown by Neira-Peña et al. (2015) [147] and Cheng et al. (2018) [159].

The main conclusion is that brain damage continues long after re-oxygenation, extending to days and/or weeks after PA, implying changes in metabolism, redox homeostasis, and suppression and/or over-activation of gene expression.

### 4.6. A Schematic Summary of the Metabolic Cascade Elicited by PA

Figure 3 summarizes the consequences of oxidative stress and cell death triggered by PA in the hippocampus of a rat neonate. At P1 (A), the effect of PA is shown as promoting a down-regulation of TIGAR expression, favoring glycolysis, to compensate for the reduced ATP synthesis induced by PA. Consequently, only basal NADPH levels are available, together with the reduced enzymatic activity of GR, resulting in accumulation of GSSG, without any changes in GSH levels. Therefore, a decrease of glutathione-dependent enzymes is observed, including GPx, resulting in H_2_O_2_ accumulation, favored by reduced catalase activity. The reduced activity of antioxidant enzymes and the high GSSG:GSH ratio promote oxidative stress, causing DNA damage, DNA fragmentation, and activation of DNA repair mechanisms mediated by XRCC1 and PARP-1. At P14, after PA (B), there is an up-regulation of TIGAR expression, favoring the PPP to increase NADPH levels. Reduced enzymatic activity of GR is, however, observed, leading to accumulation of GSSG, suggesting that the NADPH produced by the PPP increases pro-oxidant pathways, such as NADPH oxidase, favoring ROS production, and inactivation of antioxidant enzymes by permanent oxidative stress, making insufficient the increased NADPH levels by the PPP. Consequently, GSH levels are reduced, as well as glutathione-dependent enzymes, such as GPx and catalase, showing reduced enzymatic activity, favoring the accumulation of H_2_O_2_. The oxidative stress produced by the reduced activity of antioxidant enzymes causes DNA fragmentation, without changes on XRCC1 expression and/or other sentinel proteins, leading to increased calpain-dependent cell death.

## 5. Vulnerability to Recurrent Metabolic Insults

We proposed that the metabolic insults occurring at birth (a first hit) prime development, increasing vulnerability to recurrent insults (secondary and tertiary hits), challenging postnatal development and maturity. In humans, PA is a risk factor for several disorders, including learning deficits and schizophrenia [160]. In rats, PA is associated with delayed cell death, and neurochemical and related behavioral deficits [40,161].

Symptoms of several mental and/or neurological diseases can emerge first in late adolescence or early adulthood, becoming increasingly severe with increased age, linked to precipitating events occurring early in life [16]. It is proposed that there are metabolic cascades triggered by early insults that prime the CNS or make the individual vulnerable to *noxiae* probably to occur along with development, leading to progressive diseases. Mitochondrial structure, function, and energy metabolism change over time, meaning that the physiology of mitochondria evolves along the life span of an individual [162], altering oxidative metabolism.

Upon delivery, oxidative stress is a permanent risk for the developing individual, enhanced by any sudden increase or decrease of metabolic conditions, associated with development itself or to environmental-dependent conditions, including malnutrition, fatigue, fever, infections, or any inflammatory-inducing injuries [163]. Oxidative stress causes an imbalance that favors the production of ROS over antioxidant defenses [164]. H_2_O_2_ levels play a pivotal role [165]. At low levels (1–10 nM), H_2_O_2_ leads to adaptive stress responses, while above 1 μM H_2_O_2_ induces inflammation, growth arrest, and cell death [163,166].

### 5.1. Organotypic Cultures

We studied the issue of vulnerability to recurrent insults with a model combining global PA with in vitro experiments, taking advantage of the features of organotypic cultures, extensively validated for the reconstruction of brain neurocircuitries [167,168,169], shown to reproduce the regional neurochemical changes induced by PA [71]. Thus, 2–3 days after delivery, asphyxiated and control pups were euthanized for preparing triple organotypic cultures, comprising substantia nigra, neostriatum, and neocortex on coverslips inserted into culture tubes (Nunc, Naperville, IL, USA) containing 750 μL of culture medium into the holes of a roller device exposing the cultures to gaseous or water phases every minute, all within the frame of a cell incubator at 35 °C and 10% CO_2_, according to Klawitter et al. (2007) [71]. At day in vitro (DIV) 18, cultures were treated with different concentrations of H_2_O_2_ (0.25–45 mM) for 18 h, to then be returned to the standard conditions, and after a 48 h recovery period (DIV 21/22), the cultures were either assayed for cell viability or for neurochemical phenotype and confocal evaluation in formalin-fixed cultures. It was found [17] that PA produced a regional-dependent energetic deficit leading to cell death, mainly affecting mesencephalic dopamine neurons. The ADP/ATP ratio was increased (>6 fold) in cultures from AS, compared to CS rats, evaluated at DIV 21–22. That increase reflected a permanent energetic deficit, as 1 mM of H_2_O_2_ increased >7-fold the ADP/ATP ratio in cultures from CS rats, and further increased the ADP/ATP ratio in cultures from AS rats (>3-fold). That effect also affected the GSSG:GSH ratio, increased by PA, and mimicked by H_2_O_2_ in the controls, also decreasing FRAP, suggesting the involvement of the mitochondrial transition pore [170]. Thus, PA implies a long-term energetic deficit, probably responsible for the long-term deficits reported by several labs [74,171].

### 5.2. Vulnerability to a Recurrent Metabolic Insult

The issue of cell vulnerability to a recurrent metabolic insult was further evaluated in the triple organotypic formalin-fixed cultures from AS and CS neonates, labeling for MAP-2 (neuronal phenotype), GFAP (astroglial phenotype), and tyrosine hydroxylase (TH, for dopamine phenotype). It was found that the number of MAP-2 (MAP-2/mm^3^) and TH (MAP-2/TH/mm^3^), but not GFAP (GFAP/mm^3^) positive cells was significantly decreased in substantia nigra of AS compared with that of CS triple cultures, further decreased by 1 mM of H_2_O_2_, also in CS cultures. The disruptive effect of 1 mM of H_2_O_2_ was stronger in cultures from AS than that from CS newborns on TH positive cells (decreased by ~70% versus ~30%), while the effect on MAP-2 positive cells was similar in both groups (decreased by ~60% versus ~50%). No PA effect was observed on MAP-2 positive cells, either in neostriatum or neocortex, under basal or 1 mM H_2_O_2_ stimulated conditions, although that insult increased the number of MAP-2/TUNEL positive cells, both in neostriatum (>4-fold in AS; 3-fold in CS) and neocortex (~3-fold in AS and CS) [17].

Thus, these experiments show that PA primes cell vulnerability in a regional-dependent manner, being the substantia nigra and dopamine neurons particularly prone to cell death.

## 6. Effect of PA on Oligodendrocyte Maturation

In the CNS of mammals, glial growth and differentiation are predominantly postnatal events, therefore vulnerable to metabolic challenges [172]. In this context, it is relevant to highlight the myelin-producing cells, the oligodendrocytes (OLs), generated from a precursor lineage, implying differentiation and maturation, especially susceptible to hypoxia [173].

The vulnerability of this lineage is a consequence of (i) neurochemical features (i.e., low antioxidant glutathione levels, high intracellular iron stores, high production of H_2_O_2_, expression of AMPA receptors lacking GluR2 subunits); (ii) a complex differentiation program, precisely regulated by intrinsic and extrinsic cues (e.g., secreted and contact-mediated factors, protein kinases and phosphatase signaling, extracellular matrix and cytoskeletal reorganization molecules, as well as transcription factors), and (iii) high metabolic rate and ATP requirements for the synthesis of large amounts of myelin [174,175,176]. Indeed, the myelin sheath is an extended and modified oligodendroglial plasma membrane, wrapped concentrically around the nerve axon [177], with structural and functional roles, providing the insulation that allows the saltatory conduction of action potential, but also providing support for growth, stability and axonal maintenance [178,179], playing an important role for neuronal survival, as well as for the establishment and consolidation of neuronal networks during neurodevelopment. Therefore, oligodendroglial damage and myelination deficiency disrupt CNS development, leading to motor, sensory and cognitive deficits, upon the affected systems [180].

### 6.1. Effect of PA on Myelination

In rodents, pre-OLs are expressed at P2, a phenotype highly vulnerable to hypoxic challenges, while myelination starts at the first week [181], with a renewal process that continues throughout adult life [182,183]. We studied the issue with the rat model of global PA, induced at a period of maximal vulnerability, investigating the consequences on myelination, focusing on telencephalic white matter (external capsule, corpus callosum, cingulum) at P1, P7, and P14.

Telencephalic tissue samples were taken from CS and AS newborns, homogenized for purifying total RNA. Gene transcripts were amplified by RT-qPCR, quantified by the ΔΔCT method, using β-actin as a housekeeping gene. We found [68] that myelin basic protein (MBP), and Olig-1 and Olig-2 mRNA levels increased along with development in CS and AS newborns, without any significant difference among the conditions, except for Olig-1 mRNA levels, a transcription factor involved in the modulation of OLs function, that under pathological conditions, is able to promote repair of demyelinating lesions [184]. At P7, Olig-1 mRNA levels increased in AS compared to the CS condition, suggesting a myelin repairing mechanism. In coronal formalin-fixed sections, myelinated fibers were detected (MBP-positive immunofluorescence) and monitored by confocal microscopy, observing a decrease of MBP-positive signals in sections from PA-exposed, versus sections from control animals. Such decrease was only observed at P7, affecting the external capsule (decreased by ~50%), corpus callosum (by ~60%), and cingulum (by ~70%) [68]. No differences were observed at P14, indicating a CNS plasticity to respond in line with the brain injury [185].

### 6.2. Effect of PA on Glial Cells in Telencephalic White Matter

While myelination is primarily driven by mature OLs phenotype, astrocytes and microglia also contribute to myelination of the developing brain. Indeed, astrocytes and microglia provide a proper environment for oligodendroglial lineage differentiation, by secreting soluble factors enhancing axonal myelination, including platelet-derived growth factor (PDGF) [186]. Astrocytes also synthesize and secrete lipids for contributing to myelin formation, modulating cholesterol and fatty acid metabolism, via sterol regulatory element binding (SREBPs) and SREBP cleavage-activating (SCAP) proteins [187]. The effect of PA on mature OLs (MBP positive cells), astrocytes (GFAP positive cells), and microglia (Ionized calcium-binding adaptor molecule 1, Iba-1 positive cells) was further investigated in the same telencephalic coronal section of AS and CS animals taken at P7. We observed that the number of MBP cells/mm^3^ was decreased in AS compared to that in CS animals (by ~50%), without any effect on astrocytes and microglia markers [68], supporting the view that OLs are the most vulnerable glial cells to hypoxia-reoxygenation injury, at least in the examined brain regions. In the external capsule and corpus callosum, TUNEL/MBP-DAPI/mm^3^ quantification was increased in AS compared to the CS condition, suggesting that PA increases OLs specific cell death, but also disrupts oligodendroglial lineage maturation, explaining the significant decrease observed in the number of OLs.

## 7. Therapeutic Strategies to Prevent the Long-Term Effect of PA

### 7.1. Hypothermia

At present, the main therapeutic option against the long-term effects of PA is based on hypothermia, which provides protection only if initiated soon after the insult [188]. No consensus on clinical protocols has been achieved to date, and the advantages and disadvantages of head or whole-body cooling are still debated, including safety and developmental considerations [189,190,191,192,193]. The main drawback is the existence of a narrow therapeutic window [8,191,192]. The rationale of hypothermia is the graded reduction of cerebral metabolism, about 5% for every 1 °C of temperature reduction [194,195]. Cooling reduces post-depolarization release of excitatory amino acids by hypoxia-ischemia, both in the newborn [196] and adult [197] subjects. In a pioneering microdialysis experiment, Urban Ungerstedt and collaborators (1998) demonstrated in newborn piglets that hypothermia started immediately after hypoxia-ischemia was associated with reduced levels of excitatory amino acids and reduced NO efflux compared to the condition without hypothermia [198], supporting evidence that glutamate antagonists prevent events occurring during the early recovery phase following hypoxia-ischemia, before failure of mitochondrial function takes place. The effect of glutamate antagonists is however evident only when associated with hypothermia [75,199]. The team led by Alistair J Gunn and Joanne O Davidson has recently reported a very authoritative opinion on the challenges of developing therapeutic strategies for neonatal encephalopathy [193]. Here, we focus, however, on therapeutic strategies investigated and/or discussed by our studies.

### 7.2. N-Acetylcysteine

GLAST (EAAT1) and GLT-1 (EAAT2) are the main glutamate transporter protein subtypes, expressed by astrocytes [200], providing a target for increasing or decreasing extracellular glutamate levels. N-acetylcysteine, a clinically established antioxidant, activates the cystine/glutamate antiporter, modulating extracellular glutamate levels [28,201]. The direction of the transport of cystine or glutamate depends upon the intra- and extracellular concentration of the respective molecules. There is clinical evidence showing that N-acetylcysteine is well-tolerated, suggesting a role for the treatment of neuropsychiatric disorders [201,202]. In the brain, N-acetylcysteine is deacetylated and oxidized to cystine, reduced back to cysteine to be taken up by the astrocytes, playing a role in the synthesis of glutathione (GSH) [203]. Thus, it is proposed that N-acetylcysteine is an option to be investigated [204].

### 7.3. Memantine as a Lead for a Neonatal Protecting Strategy

D145 (1-amino-3,5-dimethyladamantane), synthesized by Eli Lilly in 1968, better known as memantine, was first proposed as a putative anti-parkinsonian drug in the eighties, showing a profile mimicking that of d-amphetamine and apomorphine in the 6-hydroxy-dopamine rotational model (see [205,206,207]), confirmed by Philip Seeman and collaborators (2008) [208], demonstrating the action of memantine on D2 receptors. The main pharmacodynamic feature supporting a clinical application of memantine in stroke, ischemia, or neurodegenerative disorders is, however, based on the observation that memantine is a low affinity, use-dependent, NMDAR channel blocker with fast kinetics, not interfering with normal synaptic transmission, but blocking NMDARs only when over-stimulated [209,210,211], targeting extrasynaptic NMDR activity [212,213]. Memantine has been evaluated for efficacy in children with leukomalacia, while concerns about increasing constitutive apoptosis remain [214]. An important issue is whether memantine can prevent the excitotoxic cascade elicited by PA, administered alone or together with a PARP-1 inhibitor like nicotinamide, profiting of the mild hypothermia induced by memantine [206].

### 7.4. PARP-1 as a Target for Neuroprotection

PARP-1 inhibition is a target for protection following hypoxia/ischemia, based on studies showing that ischemic injuries are reduced in PARP(−/−) mice [207]. PARP-1 inhibitors with increasing degrees of potency decrease brain damage, and improve the outcome of perinatal injury [215,216]. PARP-1 inhibitors are protective under the condition of NAD^+^ and ATP depletion, but in the presence of NAD^+^ can sensitize cells to DNA damage and increase cell death [217], inducing apoptosis in rapidly dividing cells [218]. Hence, PARP-1 can act as a cell survival, as well as a cell death-inducing factor by regulating DNA repair, chromatin remodeling, and transcription.

### 7.5. The Vitamin B3 Family

The B3 vitamin family includes three forms, nicotinamide (niacinamide), niacin (nicotinic acid), and nicotinamide ribose, all of them converted within the body to NAD^+^. Nicotinamide has been shown to protect from the long-term effects on dopamine neurotransmission elicited by PA [219,220], proposed to act as a mild inhibitor of PARP-1 overactivation [221,222]. Nevertheless, nicotinamide has been challenged because of its low potency, limited cell uptake, and short cell viability [215,223]. We demonstrated that nicotinamide produced a long-lasting inhibition of PARP-1 activity measured in the brain and heart of AS and CS neonates, 1 and 8 h after delivery [224]. A single dose of nicotinamide (0.8 mmol/kg, i.p.) produced intracerebral levels of nicotinamide in the range of 20–25 μM, for longer than 6 h, with a ratio brain/peripheral levels of approximately 0.5, when comparing the dialysates obtained simultaneously with in vivo microdialysis from the periphery (subcutaneous) and brain (neostriatum) compartments [224]. That treatment was reported to prevent the effect of PA on apoptosis, working memory, and anxiety, evaluated at the age of three months in PA-exposed and control rats [67].

Lespay-Rebolledo et al. (2019) [94] reported that the nicotinamide treatment reversed the redox impairment induced by PA, acting as a precursor of NAD^+^ and NADP^+^ [225,226,227,228], preventing oxidative stress [229]. Furthermore, nicotinamide enhanced the GR/NADPH and GPx/GSH antioxidant-dependent responses, also downregulating the TIGAR-dependent shift to PPP. Nicotinamide increased GPx and catalase activity in AS animals, with the maximal effect observed at P14 [94].

The contribution of PARP-1 to the deficits elicited by PA was further investigated by using a siRNA to knock down the expression of PARP-1. We used a structurally modified siRNA, increasing its resistance and stability by complexing with gold nanorods conjugated to the peptide CLPFFD for brain targeting [230]. The compound was administered i.p. one hour after birth in a volume of 100 μL, and 24 h after evaluated for PARP-1 expression, finding a significant decrease of PARP-1 levels in mesencephalon and hippocampus, but not in the telencephalon of PA-exposed animals. Treatment with the naked siRNA had no effect on PARP-1 levels. The study confirmed the proposed region-dependent vulnerability of the brain, individual brain regions responding differently to the treatment, probably depending upon their intrinsic PARP-1 levels. The specific history of different brain regions with respect to hypoxia and/or re-oxygenation, as well as different postnatal maturational timing, primes the expression and/or PARP-1 activity.

Nicotinamide ribose has been shown to be more potent than other analogs to increase NAD^+^ levels, proposed as a treatment for several metabolic disorders [231], an issue that should be further investigated and compared to the effects produced by nicotinamide, including that of PARP-1 inhibition.

### 7.6. Mesenchymal Stem Cell Secretomes (MSC-S)

Mesenchymal Stem Cells (MSC) administration has been proposed as a therapeutic strategy for neonatal CNS diseases [232,233], with a wide range of anti-inflammatory, anti-oxidant, and pro-regenerative effects, implying paracrine and cell-to-cell mechanisms. We reported that a single-intra-cerebro-ventricular (i.c.v.) dose of MSC to PA-exposed rats prevented the loss of mature OLs and delayed myelination in telencephalic white matter [68], confirming the well-documented experimental efficacy of MSC [234], while there is concern about the translational potential of systemic MSC administration, suggesting instead the use of cell-free MSC-derive secretomes [235]. Secretomes contain several bioactive factors (e.g., cytokines, growth factors, extracellular matrix proteins, and microRNA) that can be secreted into the extracellular space, protecting against insults affecting the targeted tissue. Secretomes can be evaluated for dosage, potency, and safety, similar to conventional pharmaceutical drugs, lyophilized for storage and large-scale production. Secretomes have been proposed as an effective tool for tissue repair, based on their ability to suppress inflammatory responses, promoting wound repair and healing [236]. Furthermore, the specificity and efficacy of MSC secretome can be enhanced by preconditioning the cells with various bioactive factors [237], including the iron chelator deferoxamine (DFX) [234] and proinflammatory cytokines (TNF-α, IFN-γ ) [238]. DFX is a hypoxia-mimetic agent that increases the production of HIF-1α, which increases the release of proangiogenic, antioxidant, and anti-inflammatory molecules by MSC [239]. De Gregorio et al. (2020) [234] showed that the systemic administration of secretome derived from DFX-preconditioned MSC reduced chronic inflammation and showed neuroprotective effects in an animal diabetic polyneuropathy model. Furthermore, the secretome from TNF-α and IFN-γ preconditioned MSC displayed enhanced antioxidant and anti-inflammatory activity, characterized by increased levels of anti-inflammatory cytokines, i.e., IL-10 and TGF-β1, reducing neuroinflammation and oxidative stress induced by alcohol and nicotine consumption [238].

We have recently reported [95] that intranasal administration of a secretome derived from preconditioned MSC to asphyxia-exposed rat newborns markedly prevented oxidative stress (GSSG/GSH) in the hippocampus, increasing Nrf2 translocation and levels of its NQO1 antioxidant effector. Also, MSC-derived secretomes decreased neuroinflammation, nuclear NF-kB/p65 levels, microglial reactivity, and cleaved-caspase-3 cell death, improving the neuro-behavioral development evaluated up to two months of age [95]. Thus, intranasal administration of secretome-derived from preconditioned MSC is a novel therapeutic strategy to prevent the short- and long-term effects of PA. Nevertheless, the mechanisms involved in the secretome effects remain to be elucidated.

### 7.7. A Schematic Summary of Proposed Therapeutic Strategies

Figure 4 schematically summarizes the strategies proposed for reducing the postnatal progression of brain damage induced by PA. Proposed therapeutic strategies are indicated, together with molecular targets and mechanisms of damage.

## 8. Conclusions

The issue of the short- and long-term consequences of PA has heuristic relevance. PA implicates a long-term biological vulnerability that depends on the length and severity of the insult. PA produces a specific metabolic/energetic insult associated, first, to delay and/or interruption of autonomous breathing and, second, to re-oxygenation, a requirement for survival. PA provides a framework to address a fundamental issue affecting long-term CNS plasticity. The insult triggers a domino-like sequence of events making the developing individual vulnerable to recurrent adverse conditions, decreasing his/her coping repertoire. PA down-regulates TIGAR expression, favoring glycolysis, to compensate for the reduced ATP synthesis induced by PA, increasing NADPH levels, leading, however, to pro-oxidant pathways. An increase of NADPH oxidase levels favors ROS production, and inactivation of antioxidant enzymes, making insufficient the shift to the PPP. As a consequence, GSH levels and glutathione-dependent enzymes are decreased, also that of catalase, favoring the accumulation of H_2_O_2_, leading to DNA fragmentation, and calpain-mediated cell death. PA also impairs myelination in grey and white matter, increasing OLs specific cell death, disrupting the maturation of the oligodendroglial lineage. The observation that brain damage continues long after re-oxygenation, extending to days and/or weeks after PA, provides a therapeutic opportunity for lessening the long-term effects elicited by a metabolic insult occurring at birth. Several therapeutic strategies, including hypothermia, N-acetylcysteine, memantine, nicotinamide, and intranasally administered mesenchymal stem cell secretomes are proposed. Intranasal administration of secretome derived from preconditioned MSC, in particular, is a novel therapeutic strategy to prevent the short- and long-term effects of PA, promising a most relevant translational potential.

## Figures and Tables

**Figure 1 antioxidants-11-00074-f001:**
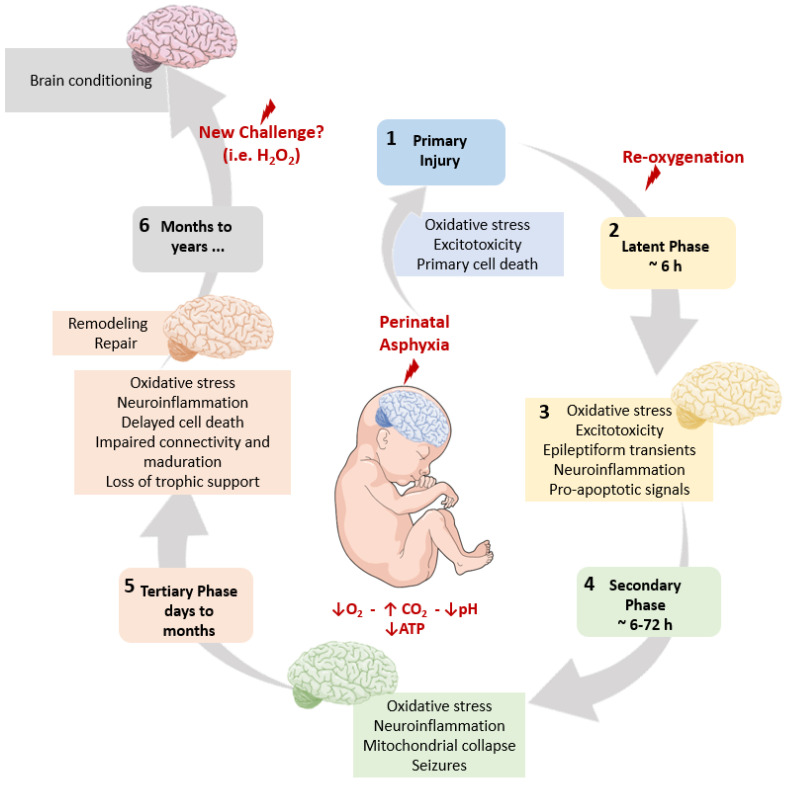
The pathophysiological cascade elicited by Perinatal Asphyxia (PA).

**Figure 2 antioxidants-11-00074-f002:**
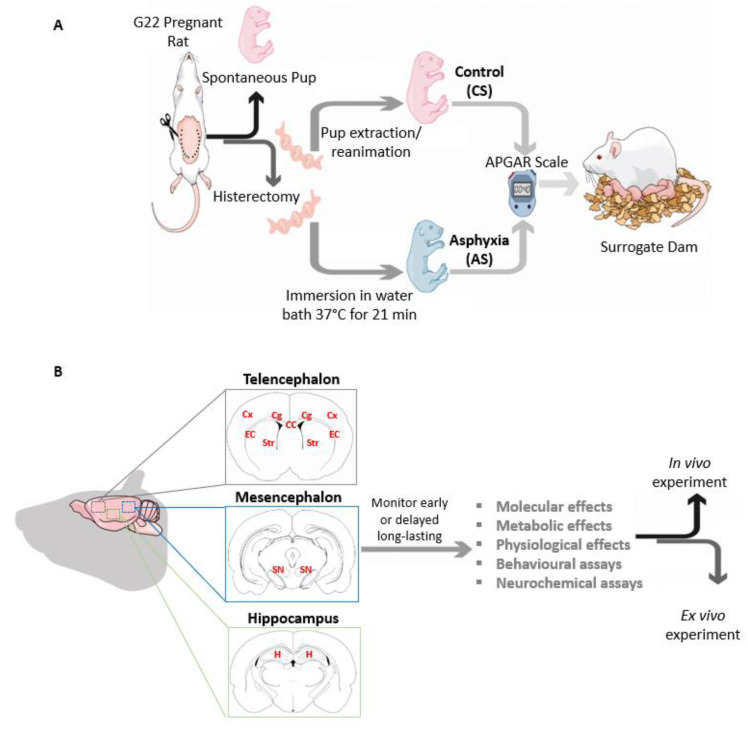
(**A**,**B**) A model of global perinatal asphyxia in rats.

**Figure 3 antioxidants-11-00074-f003:**
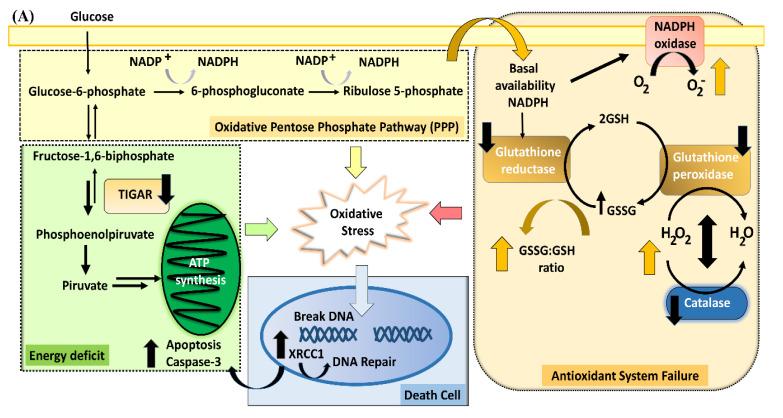
(**A**,**B**): Mechanism of oxidative stress and cell death induced by perinatal asphyxia (PA) in the hippocampus of rat neonates.

**Figure 4 antioxidants-11-00074-f004:**
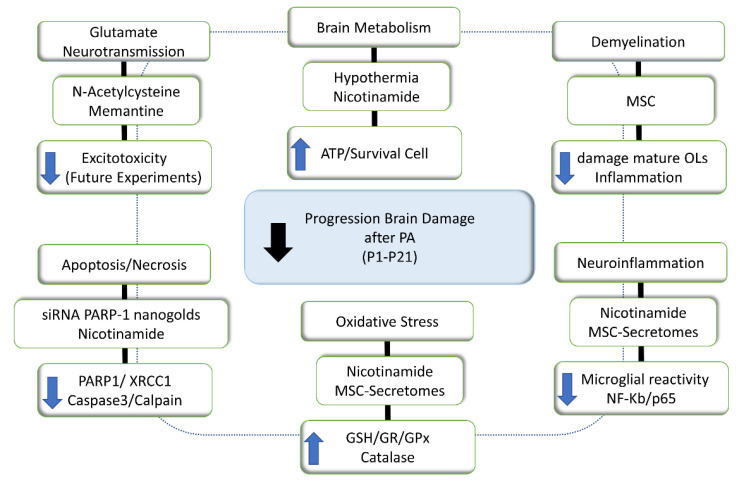
Schematic summary of therapeutic strategies.

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
