# Peer review of "Sustained Energy Deficit Following Perinatal Asphyxia: A Shift towards the Fructose-2,6-bisphosphatase (TIGAR)-Dependent Pentose Phosphate Pathway and Postnatal Development"

_antioxidants, 2021, doi:10.3390/antiox11010074_

Round 1
Reviewer 1 Report
The authors of the manuscript entitled “Sustained Energy Deficit Following Perinatal Asphyxia: A Shift towards the TIGAR-Dependent Pentose Phosphate Pathway and Postnatal Development” present a long, wide-ranging, and slightly confused description of multiple pathways associated with exposure to asphyxia in the extremely preterm-equivalent rat brain. There are some sections of the manuscript that are concise and interesting, but in general the vast majority of the content isn’t in line with the title. The manuscript is also half review paper and half a summary of the group’s own work in a single model. The abstract reads like a primary research paper, but the bulk of the paper is more like a review paper. In fact, this reads as if it is an entire doctoral thesis converted into a single manuscript. While the idea is a good one, the manuscript should be significantly shortened, focused, and more directly referenced to the clinical scenario that it is discussing.
Major comments
- As mentioned above, it is not clear what type of manuscript this is. If it is a narrative review of this topic then it must focus less on the work by this single group and more accurately cover the topic as a whole. There is plenty of supporting evidence, so I would just suggest that the authors take less of a focus on their own work.
- The model described by the group does not accurately reflect term perinatal asphyxia (PA) in humans, and this must be addressed throughout. The E22/P1 rat brain is equivalent to the extremely preterm human brain (for instance see Hamdy et al., Exp Neurol 2020), so anything seen with respect to biochemical processes after asphyxia can only be related to that population. This model is probably not relevant to term perinatal asphyxia or hypoxic-ischemic encephalopathy in humans.
- The manuscript is very hard to read. This is mainly because many of the sentences are 5-7 lines long, with multiple commas. Many of the sentences in the manuscript should be completely re-written.
- The order of the manuscript is also sometimes confusing. For instance, section 2 on perinatal asphyxia has a paragraph in the middle that is suddenly about comparing the rat to the human, and then goes back to talking about PA.
- The majority of the paper doesn’t appear to be relevant to the TIGAR-dependent shift in PPP utilisation.
- I would recommend the manuscript be cut at least in half and focused on the topic in question.
- The section on therapies is very poorly written and not well connected to the clinical scenario. For instance, standardised protocols for hypothermia (33.5C for 72h) have been in international resuscitation guidelines for more than 10 years (see the protocols for the TOBY and NICHD cooling trials). Also, see any of multiple reviews on the topic by Alistair Gunn for a full mechanistic description of how hypothermia is thought to work.
Minor comments
- The word “imply” or “implies” is used several times when perhaps “causes” or “results in” would be more appropriate.
- It is unclear how the abbreviations AS and CS are derived.
- The term “neonates” generally refers to humans. “Animals” or “newborn rats” would be better throughout.
- TIGAR isn’t defined in the title or abstract, but if the authors wish to include it here, it must be defined.
Author Response
First of all, we would like to thank the Reviewers for their insightful opinions. We have tried to follow their advice when preparing a new version of the manuscript, hoping to improve its impact.
Reviewer 1
We have carefully read the critical opinion of the Reviewer, trying to take up his/her advice for improving the quality of the paper, which is indeed a review paper, summarizing our work, using a model of global perinatal asphyxia in rats, performed at the time when the rats normally deliver, without any clamping of vessels or use of anoxic chambers.
We thank the Reviewer for recommending the insightful review by Hamdy et al. (2020), which is now introduced in the references list.
We understand the opinion of the Reviewer about the relevance of this or any other experimental model to perinatal asphyxia in humans, but we consider that that is a limitation of any in vivo or in vitro model. The models, just as models, are useful for scientific progress, while replying biochemical processes, probably relevant and conserved along species.
We have re-organize the manuscript, hoping to make clear our messages, please, see the new version of the manuscript.
Following the advice of the Reviewer, we have revised the section on the TIGAR-dependent shift in PPP utilisation, while we still believe that our results on the issue are relevant for explaining long-term metabolic deficits. The title has been modified by defining the bisphosphatase role of TIGAR.
We have tried to make clear that we do not attempt to make a full review of reported and proposed therapeutic options, while we recognise the relevant contribution that Prof. Alistar J Gunn and his collaborators have made to the field, hence including a recent review by this outstanding research group. Please, see that in the re-submitted version. We have carefully considered all the minor comments indicated by the Reviewer, modifying and/or clarifying the terms or the abbreviations, giving special consideration about the use of the term neonate or newborn to human or rodent subjects.
Reviewer 2 Report
Useful summary of data on mechanistic data related to perinatal asphyxia.
Slightly "florid" writing style, so sometimes difficult to read.
Here are some suggestions for improvement and requests for more information on the animal model (=minor revisions):
(Numbers refer to pdf line numbers)
22-23
"a risky episode, whether oxygen supply is interrupted"
to
"a risky event if oxygen supply is interrupted"
25
"marshalling "
to
"impairing"
36
"finding"
to
"showing"
53
"implying"
to
"involving"
58
"contraction-associated proteins"
to
"contraction-associated molecules"
65
"episode"
to
"event"
79
"PA is a relevant cause of death at the time of labour"
to
"PA is a relevant cause of fetal death at the time of labour"?
895
"vaginal smear, hours after a programmed mating"
to
"vaginal smear after a programmed mating"
99-101
"The issue of short- and long-term consequences of PA has heuristic relevance, since PA implicates a long-term biological vulnerability that fully depends on the severity of an insult occurring at birth, independent upon any genetic or clinical predisposition"
could be shortened to
"PA can induce long-term biological vulnerability, depending on its severity, independent of any genetic or clinical predisposition"
115
"a discover"
to
"a discovery"
208-209
"making the neonate vulnerable to metabolic insults probably to occur along life"
to
"making the neonate vulnerable to metabolic insults later in life"?
214
"nineties"
to
"1990s"
222-224
"Clinical and behavioural observations are further considered, including a first spontaneous birth, before the animal is euthanised to be subjected to a caesarean section and hysterectomised"
Do you mean
"Clinical and behavioural observations are recorded up until spontaneous delivery of the first fetus; the dam is then euthanised and subjected to caesarean section"?
Please describe euthanisation method (if anesthesia, how are dams anesthetized without anesthetizing the fetuses?)
249-250
"The handling of the pups is also important for their reception by a surrogate dam."
Please provide more information on surrogate dams
731-732
"The issue of short- and long-term consequences of PA has heuristic relevance, since PA implicates a long-term biological vulnerability that fully depends on the severity of the insult."
could be shortened to
"PA can induce long-term biological vulnerability, depending on its severity, independent of any genetic or clinical predisposition"
747-748
"provides a venue of therapeutic opportunities"
to
"implies that there are therapeutic opportunities"
Author Response
Reviewer 2
First of all, we would like to thank the Reviewers for their insightful opinions. We have tried to follow their advice when preparing a new version of the manuscript, hoping to improve its impact.
We appreciate the opinion of the Reviewer, considering all his/her suggestions, including that of making simpler our style, also modifying or replacing specific words.
Regarding “marshalling”, we have modified it to “contesting”, although we are still open for better suggestions. Perinatal asphyxia can be a cause of foetal or newborn death, depending whether occurring during labour, before, or after delivery.
We have modified the sentence referred to the relevance of perinatal asphyxia as a model of disease according with the Reviewer’s suggestion, but we keep our view on the issue, summarised at the Conclusions.
We have completed the information about the euthanasia. The dams are neck dislocated, as the use of any anaesthesia makes the model of asphyxia inviable.
Information about the surrogate dams has been enlarged. Please, see in the new version of the manuscript.
The suggestion of the Reviewer about the first sentence of the Conclusion Section has been considered, but according to our view on the issue. Also, that about “therapeutic opportunities”
We thank the Reviewer for a valuable and very positive evaluation.
Reviewer 3 Report
It was with real pleasure that I read this review called Sustained Energy Deficit Following Perinatal Asphyxia: A Shift towards the TIGAR-Dependent Pentose Phosphate Pathway and Postnatal Development. After a very well written state of the art of the consequences of perinatal asphyxia, the authors present some therapeutic strategies to prevent the deleterious effects of perinatal asphyxia.
The article is of high scientific and literary quality. The article is of high scientific and literary quality. I find it very good as it is. However some small slight modifications could improve the manuscript.
- Fig 1: “Latent Phase ~ 6h”: Wouldn't it be interesting for this element by adding that this phase corresponds to the therapeutic window? (simple suggestion).
- Line 220: Could the authors specify the strain of rat used (wistar, sprague dawley, ...)
- Line 291: The GSSG (Oxidized gluthatione ) abbreviation is not defined
- Line 298: Gluthatione should be directly replaced by GSH because it is already defined on line 292.
- The correct grammatical wording is: increase or decrease in – not of. Could the authors correct this in the all text? (line 388, 389, 488, …).
- While they are interested in perinatal asphyxia, the authors often refer to hypoxia-ischemia. The ischemic parameter is therefore added to the asphyxia. Does this have any consequences? Could the authors comment and clarify in the text?
- Part 7. The authors present therapeutic strategies to prevent the long-term effects of perinatal asphyxia. Certain strategies seem to be lacking (melatonin, maternal supplementation ...). Could the authors make a more exhaustive list (maybe under a table, so as not to lengthen the text) or else justify why they only cite these strategies?
I would just like to conclude by thanking the authors for their enormous synthesis effort and the quality of their work.
Author Response
Reviewer 3
First of all, we would like to thank the reviewers for their insightful opinions. We have tried to follow their advices when preparing a new version of the manuscript, hoping to improve its impact.
We really appreciate the positive evaluation provided by the Reviewer, thank you very much indeed, because that kind of review stimulates us to progress with our work.
Regarding Fig. 1. The Reviewer is right, but our opinion is that each phase, including the long-term metabolic deficit observed weeks after the insult provides a therapeutic opportunity, including that making the newborn vulnerable to recurrent insult. The first 6h is the present therapeutic window for hypothermia. We have modified the sentences following the Reviewer’s advice.
We have now specified the strains of the rats, Sprague-Dawley and /or Wistars rats, no seen any differences. Thank you very much for asking to specify.
Thank you very much. The proper abbreviation of glutathione is now indicated in lines 548-549 of the new version of the manuscript.
The grammatical wording indicated by the Reviewer has been considered when preparing a new version of the manuscript. Thank you very much for that.
The term global perinatal asphyxia is used for hypoxia induced at the time of delivery, without considering any vessel clamping or anoxia chambers. Ischemia is referred to models clamping one or several vessels for different times, and hypoxia-ischemia when the clamping is followed by a period in a anoxic chamber (with N and/or CO2), frequently performed at 7 day postnatal.
Thank you very much for suggesting to expand the review of therapeutic strategies to melatonin, maternal supplementation and to other stretegies. Nevertheless, we would prefer to focus on approaches we have used or discussed in our studies.
Thank you very much for your positive evaluation.
Round 2
Reviewer 1 Report
The authors have chosen to only make cursory attempts to address my concerns. As such, all my major comments remain, and this does not constitute a significant enough revision to warrant publication.
